# Interleukin-10 control of pre-miR155 maturation involves CELF2

Jeff S. J. Yoon[1,2,3☯], Mike K. Wu[1,2,3☯], Tian Hao Zhu[3], Helen Zhao[iD][4], Sylvia T. Cheung[1,2,3], Thomas C. Chamberlain[1,2,3‡], Alice L-F. Mui[iD][1,2,3‡*]

1 Immunity and Infection Research Centre, Vancouver Coastal Health Research Institute, Vancouver, Canada, 2 Department of Surgery, University of British Columbia, Vancouver, Canada, 3 Department of Biochemistry and Molecular Biology, University of British Columbia, Vancouver, Canada, 4 Department of Microbiology and Immunology, University of British Columbia, Vancouver, Canada

☯ These authors contributed equally to this work.
‡ TC and ALFM are co-senior authors.
* alice.mui@ubc.ca

## Abstract

The anti-inflammatory cytokine interleukin-10 (IL10) is essential for attenuating inflammatory responses, which includes reducing the expression of pro-inflammatory microRNA-155 (miR155) in lipopolysaccharide (LPS) activated macrophages. miR155 enhances the expression of pro-inflammatory cytokines such as TNFα and suppresses expression of anti-inflammatory molecules such as SHIP1 and SOCS1. We previously found that IL10 interfered with the maturation of pre-miR155 to miR155. To understand the mechanism by which IL10 interferes with pre-miR155 maturation we isolated proteins that associate with pre-miR155 in response to IL10 in macrophages. We identified CELF2, a member of the CUGBP, ELAV-Like Family (CELF) family of RNA binding proteins, as protein whose association with pre-miR155 increased in IL10 treated cells. CRISPR-Cas9 mediated knockdown of CELF2 impaired IL10's ability to inhibit both miR155 expression and TNFα expression.

## Introduction

MicroRNAs (miRNA) are small noncoding RNAs originally discovered in *Caenorhabditis elegans*, involved in the regulation of cell proliferation and differentiation in early embryonic development [1]. Since then hundreds of miRNAs have been identified in the human genome, as they play critical roles in post-transcriptional regulation of gene expression in diverse cell types and affiliated cellular pathways [2]. In the past decade, there is increasing attention to the role of miRNAs, and miR155 in particular, in inflammatory responses induced by pathogen-associated molecular pattern such as the bacteria cell wall component lipopolysaccharide (LPS) [3, 4].

The miR155 gene is located in an exon of B cell integration cluster, on chromosome 21 in human and chromosome 16 in mouse [5, 6]. O'Connell *et al.* found that the upregulation of miR155 increased production of the inflammatory cytokine TNFα from LPS stimulated macrophages [3]. The macrophage is one of the main white blood cells involved in both innate and

**Data Availability Statement:** The mass spectrometry proteomics data have been deposited to the ProteomeXchange Consortium via the PRIDE partner repository with the dataset identifier PXD015720.

**Funding:** These studies were supported by operating grants from the Canadian Institutes of Health Research (MOP-133415) and the Natural Science and Engineering Research Council (RGPIN-2014-05662) to ALFM. The funders had no role in study design, data collection and analysis, decision to publish, or preparation of the manuscript.

**Competing interests:** The authors have declared that no competing interests exist.

humoral immune responses [7]. Macrophage activation, which is induced by pathogen recognizing receptors, leads to phagocytic activities and the release of pro-inflammatory molecules necessary for accessing infected areas as well as attracting other myeloid cells. However, excessive release of inflammatory molecules can lead to various inflammatory diseases such as Crohn's disease and ulcerative colitis. Therefore, there are necessary checkpoints to regulate the extent of inflammation, one of which is the cytokine interleukin 10 (IL10).

IL10 was first discovered as a secreted factor produced by $Th_2$ T cells to inhibit $Th_1$ T cell -produced-cytokines, but one of its major *in vivo* actions is to deactivate the activated macrophage [8]. IL10 receptor (IL10R) signalling is best known to involve the signal transducer and activator of transcription 3 (STAT3)-dependent pathway [9–12]. However, we found that IL10R signalling also involves the Src homology 2 domain containing inositol phosphatase 1 (SHIP1) [9, 13], and that both STAT3 and SHIP1 are required for IL10 inhibition of TNFα production and miR155 levels [9, 13].

SHIP1 is itself a miR155 target gene, and LPS-stimulated elevation of miR155 reduces SHIP1 levels [14]. McCoy *et al.* showed that IL10 inhibits miR155 expression and simultaneously increases SHIP1 expression through STAT3-dependent pathway [15]. We subsequently found IL10 uses both STAT3 and SHIP1 to inhibit LPS-induced miR155, and that this occurs through inhibiting the maturation of pre-miR155 to mature miR155 [13]. Thus understanding the mechanism underlying IL10-induced pre-miR155 maturation would give insight into a key regulatory point for IL10 control of macrophage activation.

Ruggiero *et al.* studied LPS control of miR155 expression in macrophages, and implicated the KH-type splicing regulatory protein (KSRP) in the maturation of pre-miR155 to miR155 [16]. Ruggiero *et al.* co-immunoprecipitated KSRP and pre-miR155 RNA, and siRNA-mediated knock-down of KSRP impaired miR155 expression [16]. In lung epithelial cells, Bhattacharyya *et al.* reported that KSRP promoted miR155 expression while tristetraprolin (TTP) inhibited miR155 levels [17]. Whether IL10 inhibition of miR155 maturation occurs through regulation of either KSRP or TTP is not known.

In this study we aimed to identify proteins which associate with pre-miR155 in macrophages in response to IL10 in order to gain insight into the proteins that regulate pre-miR155 maturation. Our approach was to transfect biotinylated pre-miR155 oligonucleotides into a macrophage cell line, followed by isolation of pre-miR155 associated proteins using streptavidin magnetic beads. Proteins in these RNA pulldown samples were identified by mass spectrometry. One of these, CELF2 (CUGBP Elav-like family member 2) we now report as a potential regulator of pre-miR155 maturation. IL10's ability to inhibit maturation of pre-miR155 and expression of TNFα were impaired in cells in which CELF2 had been knocked down.

## Materials and methods

### Cells

RAW264.7 cells were obtained from the American Type Culture Collection and maintained in Roswell Park Memorial Institute 1640 medium (RPMI-1640) (HyClone, Logan, Utah) supplemented with 9% fetal bovine serum (FBS) (HyClone, Logan, Utah). Transduced cells were selected with 10 μg/ml blasticidin.

### Constructs

Lentiviral expression vectors for the doxycycline inducible CRISPR-Cas9 and sgRNA were purchased from Addgene (Lenti-iCas9-neo #85400; pLX-sgRNA #50662 [18, 19]). Using CRISPR Gold online tool [20], guide RNA sequence targeting CELF2 gene has been designed: CELF2 target sequence (5' GCACTTACCGCCAGGTACTG). The target sequence was cloned

into pLX-sgRNA vector by first using overlap-extension PCR to generate fragments of CELF2-sgRNA-specific inserts. The PCR was performed with Phusion polymerase (F-549L, ThermoFisher Scientific, Nepean, ON) and the PCR product was gel purified and extracted using phenol:chloroform:isoamyl (25:24:1) alcohol (15593–049, ThermoFisher, Nepean, ON) to remove any template pLX-sgRNA vector and Phusion polymerase that can interfere in generating correct clones in subsequent steps. The purified PCR product was then used as a template for another round of PCR reaction to generate the full length of CELF2-sgRNA-specific insert. Following the gel purification and phenol chloroform extraction, the second PCR product was digested with XhoI and NheI restriction digest enzymes (New England Biolabs, Ipswich, MA) and ligated into digested pLX-sgRNA. CELF2-specific insert containing pLX-sgRNA vector was transfected into chemically competent Stbl3 cells and the colonies were selected using ampicillin. The sequences of the vector from the ampicillin resistant colonies were confirmed by sequencing. The virus containing Lenti-iCas9 neo or CELF2-plx-sgRNA vector were prepared by mixing the vector with the packaging plasmid R8.9 and VSVG and transfecting into HEK293T cells to produce virus. The lentivirus in the media was collected, concentrated by ultracentrifugation and incubated with RAW264.7 cells in the presence of 8 μg/ml protamine sulfate. RAW264.7 cells expressing Cas9 were constructed by infection with Lenti-iCas9 derived lentiviruses. Transduced cells were maintained in 2 mg/ml neomycin in 9% FBS/RPMI-1640. RAW264.7 cells expressing Cas9 were further infected with CELF2-pLX-sgRNA viruses and selected using 10 μg/ml blasticidin. To induce the expression of Cas9, 2 μg/ml doxycycline was added to the culture media for up to 96 hours prior to the cells being used for experiments.

### RNA-oligonucleotide transfection

RAW264.7 cells were seeded at $1.2 \times 10^6$ cells per well in 6-well tissue culture plates 1 day prior to transfection. Biotinylated pre-miR155 oligonucleotides (Biotin-UAAUUGUGAUAGGGGUUU UGGCCUCUGACUGACUCCUACCUGUUA) and TNFα ARE (Biotin-UUAUUAUUUAUUAUUUAU UUAUUAUUUAUUUAUUU) were obtained from Invitrogen Life Technologies (ThermoFisher Scientific, Nepean, ON). RNA oligonucleotide was prepared in Opti-MEM (ThermoFisher Scientific, Nepean, ON) at a concentration of 0.86 μM. Lipofectamine-2000 (ThermoFisher Scientific, Nepean, ON) was diluted in Opti-MEM at 1:25 volume/volume (v/v) ratio, and was mixed with RNA oligonucleotide-Opti-MEM solution at 1:1 (v/v) ratio and incubated at room temperature for 20 minutes to allow the formation of lipofectamine-oligonucleotide complexes. After incubation, the lipofectamine-oligonucleotide solution was further diluted with 9% RPMI at 1:1.25 (v/v) ratio and 675 μl of the solution were added to each wells with cells and incubated for 6 hours in 37°C, 5% $CO_2$ chamber. After 6 hours, the solution in the well was replaced with 1 ml of 9% FBS/RPMI to allow cells to recover overnight.

### Cell stimulation

Following the 9% FBS/RPMI overnight incubation, RAW264.7 cells were stimulated with 10 ng/ml of LPS (*Escherichia coli* serotype 0111:B4; MilliporeSigma, Oakville, ON) or 10 ng/ml LPS with 100 ng/ml IL10 for three hours in 37°C, 5% $CO_2$ chamber.

### Lysate collection

Following stimulation, the media was removed and cells are incubated with cold (4°C) Phosphate Buffered Saline (PBS, ThermoFisher Scientific, Nepean, ON) for 2 min. The PBS was removed, and cells were then lysed by the addition to each well of 600 μl of Protein Solubilization Buffer (PSB, 50 mM HEPES, 100 mM NaF, 10 mM NaPPi, 2 mM NaVO$_4$, 2 mM

NaMoO$_4$, 4 mM EDTA) containing 0.125% nonyl phenoxypolyethoxylthanol (NP-40), protease inhibitor cocktail (MilliporeSigma, Oakville, ON), and 0.5 mM Tris(2-carboxyethyl)phosphine (TCEP, Soltec Ventures, Beverly, MA). Cell lysates were collected with a cell scraper and rotated at 4˚C for 30 minutes, followed by centrifugation for 20 minutes at 12000 rpm, 4˚C.

## RNA pulldown assay

Supernatants from the centrifugation were added to Streptavidin Magnetic beads (MilliporeSigma, Oakville, ON) in 1.5 ml Eppendorf tubes and incubated for 90 minutes at 4˚C on a Nutator mixer. The tubes were then briefly centrifuged at 5000 rpm to bring the fluid to the bottom of the tubes, and magnetic beads immobilized using a magnetic tube stand (ThermoFisher Scientific, Nepean, ON). Lysates were removed, the beads resuspended with 900 μl of wash buffer (0.1% Tween-20 containing PSB) and rocked for 5 min at 4˚C on a Nutator. Beads are immobilized as before andwashed two more times. Proteins were eluted either by boiling in 2 X SDS-PAGE sample buffer (0.125 M Tris, pH 6.8, 5% 2-mercaptoethanol bromophenol blue, 13.5% glycerol, 4.5% SDS) for immunoblot analysis, or by incubation with high salt phosphate buffer (0.0067 M PO$_4$, pH 7.0, 1.5 M NaCl, 0.02% Tween-20) for 15 minutes on a shaker at 1500 rpm at room temperature for mass spectrometry analysis. The NaCl eluted proteins were then supplemented with 4X SDS-PAGE sample buffer to a final of 2 X SDS-PAGE sample buffer, boiled 5 min, frozen and sent to the UBC Proteomic Facility for processing and mass spectrometric identification of peptides.

## Immunoblot analysis

The proteins were separated by 10% SDS-PAGE, followed by electroblotting onto polyvinylidene fluoride (PVDF) membrane (MilliporeSigma, Oakville, ON). The membranes were blocked in 3% bovine serum albumin (BSA), then probed with the following primary antibodies overnight: 1:1000 KSRP (ab140648, Abcam, Toronto, ON), 0.5 μg/ml TTP (N-terminal, T5327, MilliporeSigma, Oakville, ON), 1:1000 pSTAT3 Tyr705 (3E2) (9138, Cell Signaling), 0.1 μg/ml Actin (A2066, MilliporeSigma, Oakville, ON), and 1 μg/ml CELF2 (Sc-47731, Santa Cruz, Dallas, TX). The membranes were washed in Tris-Buffered Saline containing 0.05% Tween-20 (TBST), incubated with with either Alexa Fluor® 660 anti-mouse IgG or Alexa Fluor® 680 anti-rabbit IgG, and imaged using a LI-COR Odyssey Imager.

## Blue-silver colloidal coomassie staining

The proteins were separated by 10% SDS-PAGE, followed by incubation in fixative solution (40% ethanol and 10% acetic acid) for 4 hours. After washing the gel 3 times with double-distilled water, 5–10 minutes each wash, the gel was incubated in blue-silver colloidal coomassie stain composed of 0.12% Coomassie G250 (ThermoFisher Scientific, Nepean, ON), 10% ammonium sulphate, 10% o-phosphoric acid, and 20% methanol overnight. The gel was washed 7 times with double-distilled water; 10–20 minutes each wash, and imaged using a LI-COR Odyssey Imager.

## Mass spectrometry

The samples were run on 10% SDS PAGE gel, ran only a short distance into the resolving gel, visualized by colloidal coomassie [21], and in gel digested [22] using trypsin (Promega, USA). The resulting peptides were cleaned using solid phase extraction on C-18 STop And Go Extraction (STAGE) Tips [23] and the peptides were analyzed on quadruple-time of flight mass spectrometer (Bruker Impact II) as [24]. The mass spectrometry data was searched on

MaxQuant version 1.5.3.30 [25], against uniprot's *Mus musculus* protein sequences (50,829 entries) plus common contaminant sequences (245 entries). The search parameters remained as default settings except for enabling match-between-run and label-free quantification features.

## RNA extraction and real time PCR

Cells were seeded at 3.0 x $10^5$ cells per well in a 24-well tissue culture plate and allowed to adhere overnight. Media was changed the next day 1 hour prior to stimulation. Cells were stimulated with 1 ng/ml LPS +/- (0.1 or 1 ng/ml) IL10 for 4 hours. Triplicate wells were used for each stimulation condition. Total RNA was extracted using Tri-Reagent (T9424, Millipore-Sigma, Oakville, ON) according to manufacturer's instructions. 1–3 μg of RNA was treated with RNase free DNase I (cat# 04 716 728 001, MilliporeSigma, Oakville, ON) for 20 minutes at 37˚C followed by addition of 0.1 M EDTA to a final concentration of 8 mM and incubation at 75˚C for 10 minutes to inactivate RNase free DNaseI. 20 ng of DNase I treated RNA was used to generate cDNA from miR155 and small nucleolar RNA MBII-202 (snoRNA202) using miRNA TaqMan Reverse Transcription Kit (cat# 4366597, ThermoFisher Scientific, Nepean, ON), MultiScribe™ reverse transcriptase (cat# 4319983, ThermoFisher Scientific, Nepean, ON) and miR155 (cat# 002571) and snoRNA202 (cat# 001232, ThermoFisher Scientific, Nepean, ON) probes according to the manufacturer's instructions. The miR155 and snoRNA202 cDNA was used with TaqMan fast advanced master mix (cat# 4444557, ThermoFisher Scientific, Nepean, ON) for real-time PCR and the expression of miRNA was measured with StepOne Plus™ (Invitrogen, Burlington, ON). The miRNA levels were quantified using the comparative $C_T$ method with snoRNA202 used as the normalization control.

## Measurement of TNFα production

Cells were seeded at 1.5 x $10^4$ cells per well in a 96-well tissue culture plate and allowed to adhere overnight. Media was changed the next day 1 hour prior to stimulation. Cells were stimulated with 1 ng/ml LPS +/- IL10 (0.5 or 10 ng/ml) for 1 hour. Triplicate wells were used for each stimulation condition. Supernatant was collected and secreted TNFα protein levels were measured using a BD OptEIA Mouse TNFα Enzyme-Linked Immunosorbent Assay (ELISA) kit (BD Biosciences, Mississauga, ON).

## Statistical analysis

Quantification of band intensities in immunoblots was performed using LI-COR Odyssey imaging system and Image Studio™ Lite software (LI-COR Biosciences, Lincoln, NE). Graph-Pad Prism 6 (GraphPad Software Inc., La Jolla, CA) was used to perform all statistical analyses. Statistical details can be found in figure legends. Values are presented as means ± standard deviations. Unpaired Student's *t* tests were used where appropriate to generate two-tailed P values. Two-way ANOVA was performed where required with appropriate multiple comparisons tests. Differences were considered significant when $p \leq 0.05$.

## Ethics statement

The cell culture experiments are done in accordance with UBC Biosafety requirements.

## Results

### Pre-miR155 associates with Tristerapolin (TTP) in response to IL10

Previous research had suggested that two RNA binding proteins might associate with pre-miR155. Ruggiero *et al.* reported pre-miR155 could be immunoprecipitated with the KH-type

splicing regulatory protein (KSRP) [16]. Bhattacharyya *et al.* concluded that KSRP promoted miR155 expression while tristetraprolin (TTP) inhibited miR155 levels [17]. Both of these RNA binding proteins were first described as ones which bind to the AU-rich element (ARE) in the 3'-untranslated region in the TNFα mRNA [26, 27] and the TNFα ARE is required for IL10 control of TNFα mRNA expression [9, 28]. To confirm our ability to pull down proteins that associate with biotinylated pre-miR155, we transfected RAW264.7 cells with biotinylated oligonucleotides corresponding to either pre-miR155, or the TNFα ARE as a positive control. One day after transfection, cells were stimulated with LPS and LPS + IL10 for 3 hours, cell lysates generated, and the biotinylated RNA-oligonucleotides/associated proteins isolated using streptavidin magnetic beads. We then probed these pulldown samples for the presence of KSRP and TTP.

As shown in Fig 1A, the amount of KSRP protein in the total cell lysates remained constant regardless of the cell stimulation. Fig 1B shows that KSRP bound better to RNA oligonucleotides of pre-miR155 than the TNFα ARE. The amount of KSRP bound remained the same regardless of stimulation, suggesting that if KSRP regulates pre-miR155 processing or TNFα translation, it does so without any changes in the amount of KSRP associated with either of these RNA elements.

The amount of TTP protein in the cell lysates increased slightly in the LPS treated samples, and significantly in the LPS + IL10 treated cells (Fig 1A), a response similar to previously reported [29, 30]. Because of the significant change of TTP protein levels in the starting cell lysates, we normalized the TTP in the pulldowns to either (Fig 1C) TTP (whose abundance changes in the LPS and LPS + IL10 cell lysates) or (Fig 1D) α-actin (which stays the same in all stimulation conditions) in the cell lysate. This allows us to consider both the relative proportion of cellular TTP (Fig 1C) and the total amount TTP (Fig 1D) bound to the RNA oligonucleotides. For TNFα ARE, the absolute amount of TTP bound increased in the LPS + IL10 samples (Fig 1D) but the relative amount (Fig 1C) stayed the same. This suggests that the binding of TTP protein to the TNAα ARE depends only on the cellular concentration of TTP protein. When more TTP protein is present, more binds to the TNFα ARE.

In contrast, for pre-miR155, both the relative (p < 0.01, Fig 1C) and absolute (p < 0.0001, Fig 1D) amount of TTP bound increases significantly in the LPS + IL10 samples as compared to either the LPS or the unstimulated samples. This suggests that the TTP molecules which bind to pre-miR155 may have undergone a post-translational modification that increases its affinity for pre-miR155.

## Identification of CELF2 as a pre-miR155 interacting protein

Having confirmed that we can detect RNA binding proteins using this transfection and pull-down method, we then scaled up the number of cells used in order to isolate proteins for mass spectrometry-based identification of potentially new pre-miR155 associated proteins. A portion of each pulldown was run on an SDS-gel, and proteins visualized with blue-silver colloidal coomassie stain (Fig 2). Several bands were detected which are unique to either the TNFα ARE or pre-miR155 pulldowns. The remainder of the samples were sent for tandem mass spectrometry. Proteomic data from mass spectrometry was analyzed with MaxQuant [31]. The number of peptides pulled down by pre-miR155 under unstimulated, LPS, and LPS + IL10 stimulated conditions for each uniquely identified protein were compared based on their respective label-free quantification intensity values (see Material and Methods). A protein called CELF2 was identified to be enriched in the LPS + IL10 pre-miR155 pulldown sample. The molecular weight of CELF2 is around 50 kDa which is similar to one of the bands unique to the pre-miR155 pulldown detected on the Coomassie gel (white box, Fig 2).

**(A)**   Cell Lysate

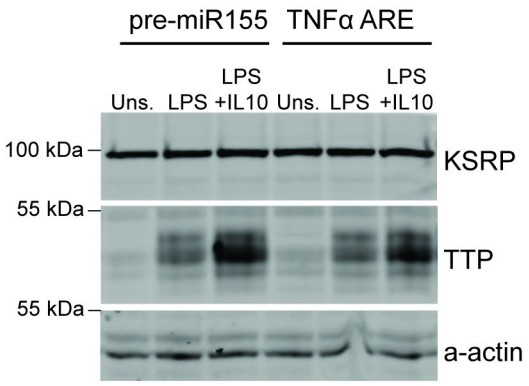
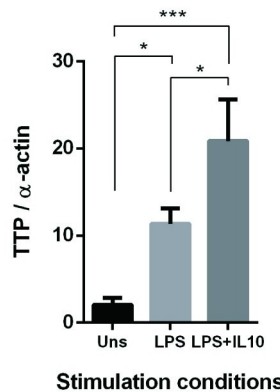
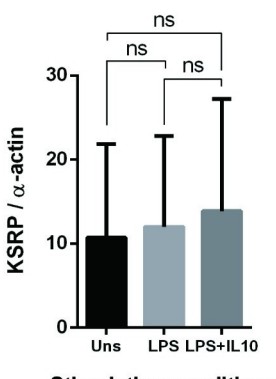

**(B)**   Pulldown

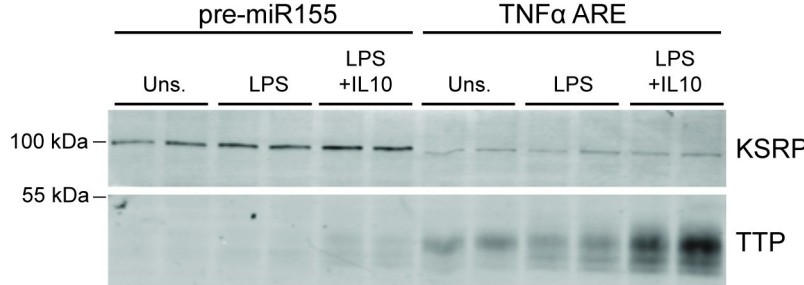

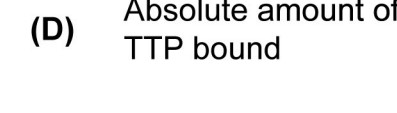

**(C)**   Amount of TTP bound relative to total TTP in cell lysate

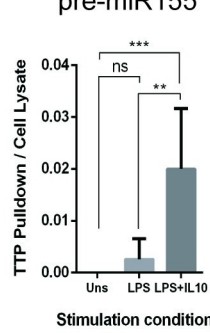
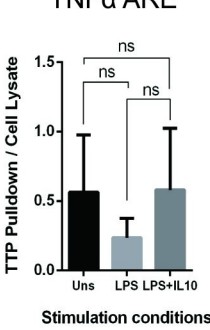

**(D)**   Absolute amount of TTP bound

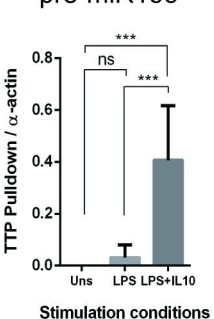
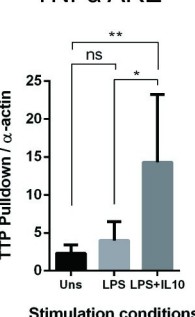

**Fig 1. Validation of RNA pulldown technique by observing known TNFα ARE interaction proteins.** RAW264.7 cells were transfected with indicated biotinylated RNA-oligonucleotides and stimulated with LPS or LPS + IL10 for 3 hours prior to collecting pulldown samples. Expression levels of proteins interacting with TNFα ARE or pre-miR155 oligonucleotides were determined by immunoblotting. (A) Immunoblot of cell lysate. Data plotted shows TTP and KSRP protein band intensities normalized to α-actin. (B) Immunoblot of the pulldown samples. Data plotted shows TTP protein band intensities for each pulldown normalized to either TTP protein (C) or α-actin (D) in total cell lysates. Data represents three independent experiments with significance between treatments calculated by One-Way ANOVA with Tukey's correction, *** $p < 0.001$, ** $p < 0.01$, * $p < 0.05$.

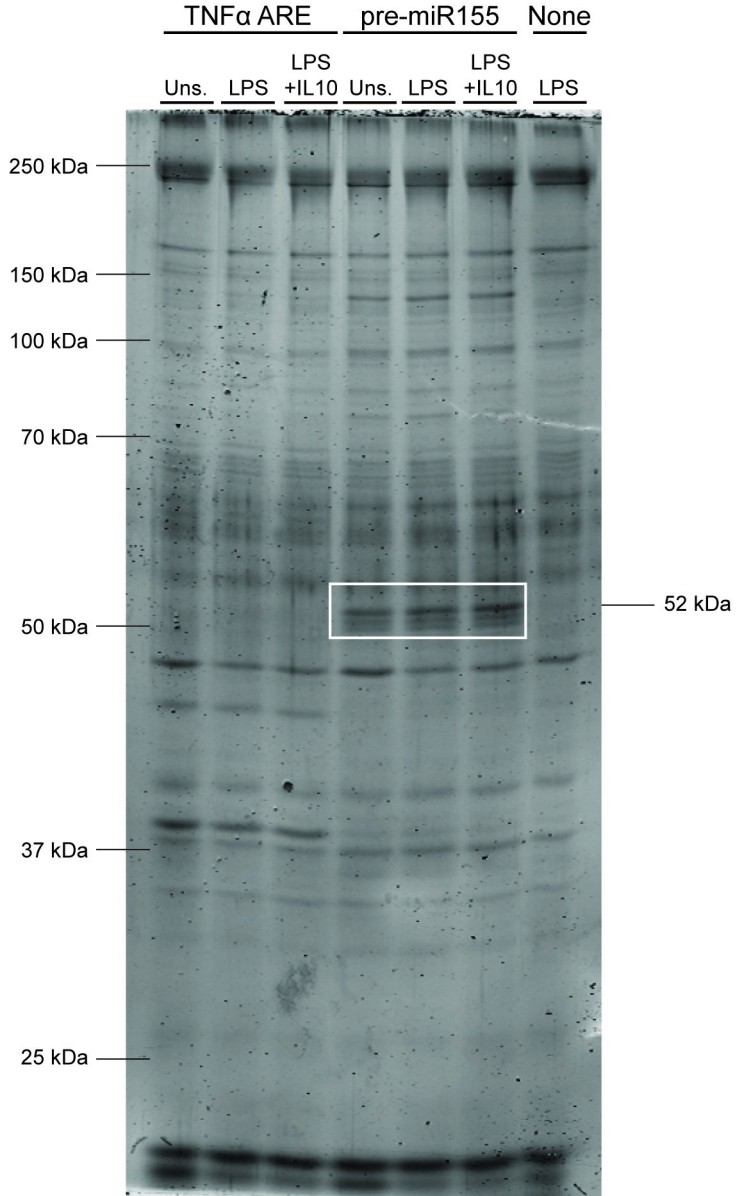

**Fig 2. Identification of proteins interacting with pre-miR155 oligonucleotides.** RAW264.7 cells were transfected with indicated biotinylated RNA-oligonucleotides and stimulated LPS +/- IL10 for 3 hours prior to collecting pulldown samples. The abundance of proteins interacting with pre-miR155 oligonucleotides were determined by blue-silver colloidal staining. The 52 kDa protein inside the white box is the protein speculated to be CELF2.

To confirm the mass spectrometry data, we probed immunoblots of the pre-miR155 pull-downs with an antibody to CELF2. Fig 3A shows lysates from cells treated with LPS +/- IL10 may express slightly more CELF2 than unstimulated cells. Fig 3B shows CELF2 binds to pre-miR155 but not TNFα ARE, similar to the behavior of the 52 kDa in the Coomassie gel (Fig 2). There was more CELF2 in LPS + IL10 miR155 pulldown samples than in either the LPS or unstimulated pre-miR155 pulldown samples regardless of whether the CELF2 protein in the pulldown is normalized to the amount of CELF2 (Fig 3C) or actin (Fig 3D) in the starting cell lysate.

**(A)** Cell Lysate

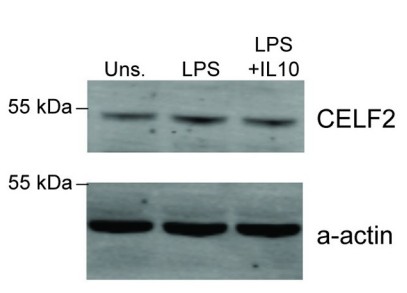

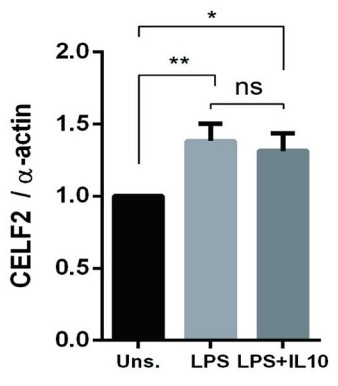

**(B)** Pulldown

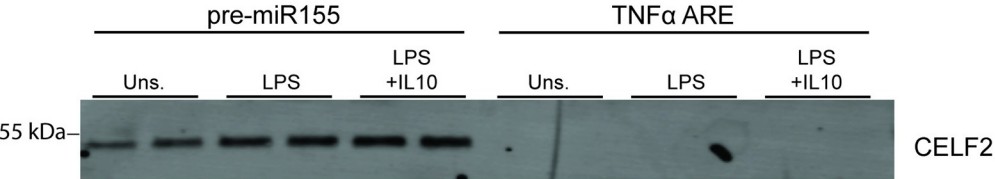

**(C)** Amount of CELF2 bound relative to total CELF2 in cell lysate

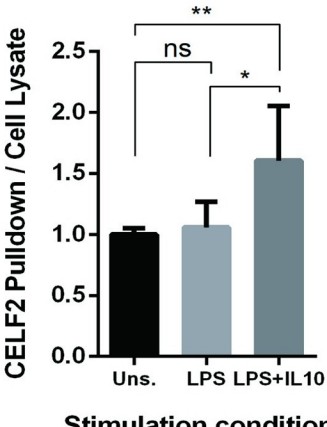

**(D)** Absolute amount of CELF2 bound

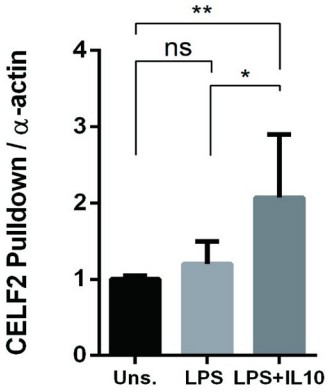

**Fig 3. Increased interaction of CELF2 protein to pre-miR155 in response to IL10 stimulation.** RAW264.7 cells were transfected with indicated biotinylated RNA-oligonucleotides and stimulated with LPS +/- IL10 for 3 hours prior to collecting pulldown samples. Expression levels of proteins interacting with TNFα ARE or pre-miR155 oligonucleotides were determined by immunoblotting. (A) Immunoblot of cell lysate. Data plotted represents CELF2 protein band intensities in cell lysate normalized to α–actin in cell lysate, relative to unstimulated samples. (B) Immunoblot of pulldown samples. Data plotted represent CELF2 protein band intensities in pulldown normalized to CELF2 protein levels (C) or α-actin (D) in total cell lysate, relative to unstimulated samples. Data represents three independent experiments with significance between treatments were calculated by One-Way ANOVA with Tukey's correction, $^{**}$ $p < 0.01$, $^{*}$ $p < 0.05$.

## The role of CELF2 in IL10 regulation of miR155

To investigate the role of CELF2 in IL10 inhibition of miR155, we used CRISPR-Cas9 to knockdown CELF2 in RAW264.7 cells as described in Materials and Methods. As shown in Fig 4A, CELF2 protein level is greatly reduced in the CELF2 KD cells compared to the non-CELF2 targeted RAW264.7 cells. We stimulated these cells with LPS +/- IL10 for 4 hours to allow miR155 induction [13], isolated RNA, and determined the level of miR155 in these samples by qPCR. In Fig 4B, the miR155 data for each cell type were normalized to the LPS only sample of each cell type in order to look at the ability of IL10 to inhibit miR155 levels within each cell type. Fig 4B shows LPS induced miRNA in both CELF2 and non-target knockdown cells, but 0.1 ng/ml IL10 inhibits miR155 expression only in the non-target knockdown cells. 1 ng/ml IL10 did slightly reduce miR155 expression in the CELF2 knockdown cells, but the degree of inhibition is significantly less than the inhibition in the non-target knockdown cells ($p < 0.05$). In Fig 4C, all the miR155 data for both cell types were normalized to the LPS only of the non-target cell in order to compare the miR155 levels between the two cell types. Fig 4C shows that CELF2 knockdown resulted in greater miR155 expression, suggesting that CELF2 is a negative regulator of miR155.

Ruggiero *et al.* reported that miR155 deficiency increases inflammatory cytokine production in response to LPS [16]. So we next examined IL10's ability to inhibit TNFα expression in the CELF2 knockdown cells. Cells were stimulated with LPS +/- IL10 for 1 hour, and TNFα levels in the cell supernatants quantified by ELISA. As shown in Fig 5, the LPS alone stimulated TNFα levels in the CELF2 knockdown cells were indeed about 30% higher than in the non-target knockdown cells ($p < 0.05$). Furthermore, 0.5 ng/ml IL10 inhibited TNFα expression in the non-target knockdown cells, but could not inhibit TNFα expression in the CELF2 knockdown cells.

## Discussion

CELF2 belongs to the CUG-BP, Elav-like family (CELF) and has been shown to possess numerous RNA regulatory activities such as mRNA binding, RNA editing, translation inhibition and alternative splicing in different tissues such as neurons, heart, muscle and kidney [32–36]. In this study, we report that CELF2 contributes to IL10 induced expression of miR155 in macrophages.

Other RNA binding proteins have been described to regulate both mRNAs and miRNAs [37]. KSRP [38] and TTP [27] were first described as proteins that bind to the TNFα 3'UTR ARE element, where they control mRNA stability and translational silencing in part by recruiting other proteins to assemble at the ARE [39]. KSRP was subsequently implicated in the processing of the Let-7 miRNA in C elegans [40], and Ruggiero *et al.* showed using siRNA knockdown studies that KSRP was required for LPS induction of miR155 in macrophages [16]. We found that the amount of KSRP associating with pre-miR155 remains unchanged in macrophages stimulated with LPS or IL10 as compared to untreated cells (Fig 1). Since the amount of KSRP binding to pre-miR155 doesn't change, the functional activity of KSRP must change in LPS + IL10 treated cells. KSRP has been reported to undergo changes in phosphorylation state [41] and this modification may change its ability to support processing of pre-miR155 to miR155 by DICER [16].

As expected from previous studies [29, 30], the levels of TTP in cells increases slightly in LPS stimulated cells and greatly in LPS + IL10 stimulated cells (Fig 1). TTP is best characterized for its binding to the TNFα ARE and inhibits translation [27], but Bhattacharyya *et al.* reported that TTP represses the expression of miR155 in lung epithelial cells [42]. TTP assembles at the TNFα ARE in a complex consisting of other RNA binding proteins (AUF1, and

**(A)** CELF2 expression in knockdown cells

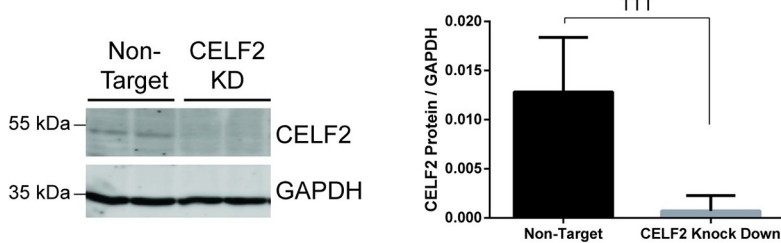

**(B)** miR155 normalized to each cell's own LPS stimulated sample

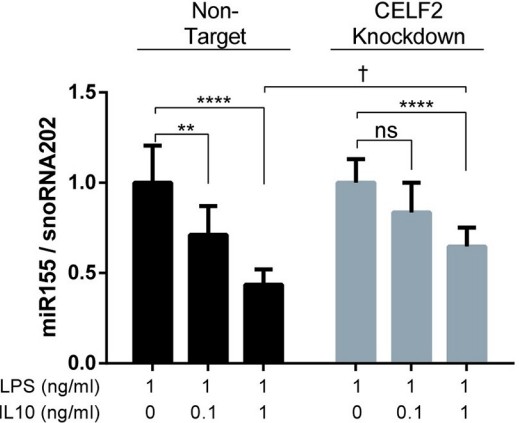

**(C)** miR155 normalized to LPS stimulated sample of the non-target cell

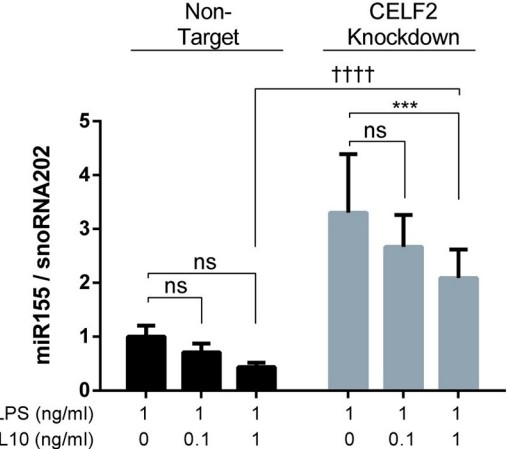

**Fig 4. Knockdown of CELF2 impairs IL10 inhibition of miR155 and increase basal level of miR155 in response to LPS.** RAW264.7/Cas9 cells transduced with CELF2 sgRNA or non-target sgRNA were treated with 2 μg/ml doxycycline to induce knockdown of CELF2 protein. (A) CELF2 protein expression was determined by immunoblotting. Data plotted represents CELF2 protein band intensity normalized to GAPDH in three independent experiments (unpaired Student's $t$ test). (B) CELF2 or non-target knockdown cells were stimulated with 1 ng/ml LPS ± indicated concentration of IL10 for 4 hours prior to total RNA extraction. Expression level of miR155 was determined by real-time PCR and normalized to snoRNA202 levels. Data plotted represents the expression miR155 relative normalized to each cell's own LPS stimulated sample or (C) qPCR data from B, but with all samples normalized to the LPS stimulated sample of the non-target cell. (Two-Way ANOVA with Tukey's correction. The asterisk (*) were used for comparison within the same cell type and the dagger (†) were used for comparison between the different cell type, **** $p < 0.0001$, *** $p < 0.001$, ** $p < 0.01$, * $p < 0.05$, † $p < 0.05$, †††† $p < 0.0001$).

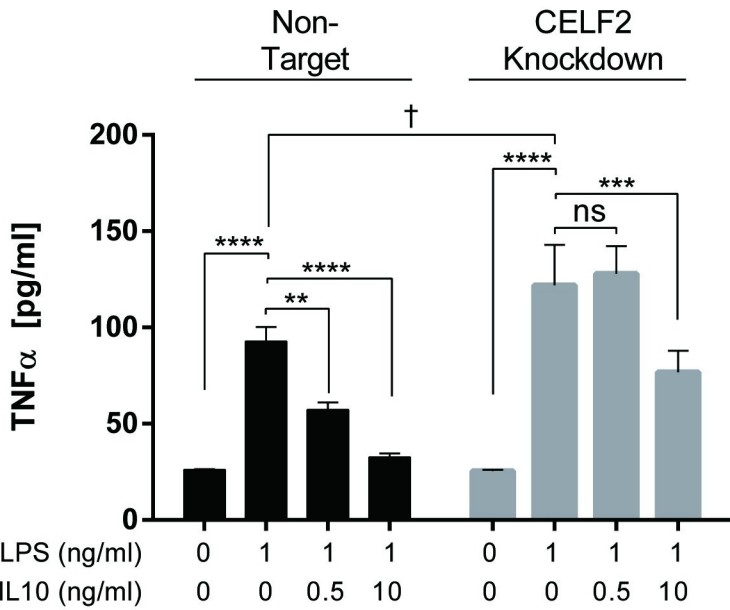

**Fig 5. Knockdown of CELF2 impairs IL10 inhibition of TNFα expression.** RAW264.7/Cas9 cells transduced with CELF2 sgRNA or non-target sgRNA were treated with 2 μg/ml doxycycline to induce knockdown of CELF2 protein. CELF2 or non-target knockdown cells were stimulated with 1 ng/ml LPS ± indicated concentration of IL10 for 1 hour prior to collecting the supernatant. The level of TNFα was determined by ELISA. (Two-Way ANOVA with Tukey's correction. The asterisk (*) were used for comparison within the same cell type and the dagger (†) were used for comparison between the different cell types, **** $p < 0.0001$, *** $p < 0.001$, ** $p < 0.01$, * $p < 0.05$, † $p < 0.05$).

TIAR) and miRNAs (miR-221, miR-579, and miR-125b) [39]. This complex recruits the miR-ISC repressor complex that can target the miRNA for degradation or inhibit translational initiation [39]. As seen in Fig 1, more TTP binds to the TNFα ARE than to the pre-miR155. Since the total amount of TTP differs in the different cell samples, we normalized the amount of TTP bound to the RNA-oligonucleotides to the total TTP in the starting cell lysate in order to determine the proportion of cellular TTP protein binding to the RNAs. The proportion of cellular TTP bound to the TNFα ARE remains the same in all stimulation conditions suggesting that this association is controlled mainly by the concentration of TTP in the cell. In contrast, the proportion of total cellular TTP bound to the pre-miR155 increased in the LPS + IL10 treated cells. This increase in the relative amount of cellular TTP bound suggests that the molecules of TTP which bind to pre-miR155 may differ from those which do not. For instance TTP can undergo phosphorylation [43]. Perhaps the increase in TTP binding to pre-miR155 is due to some post-translational modification on TTP induced by the LPS + IL10 stimulation of the cells. The role of TTP in IL10-induced translational silencing of TNFα mRNA is well established [30, 39, 44]. The potential role of TTP in controlling pre-miR155 processing to miR155 remains to be determined.

We took an unbiased approach to identifying proteins that associate with pre-miR155 by subjecting the pre-miR155 RNA pulldown samples to mass spectroscopic analysis. We found KSRP and TTP as well as a number of proteins not previously described to bind miR155 or miRNAs. We focused first on CELF2. CELF2 is an RNA binding protein reported to regulate mRNA splicing, stability, translation [45, 46] and observed to bind to ARE regions of certain mRNA [47]. We confirmed by immunoblot analysis that CELF2 does bind pre-miR155 (Fig 3). Notably, the CELF2 bound to pre-miR155 better than it binds to the TNFα ARE. We normalized the amount of CELF2 bound to pre-miR155 to the total cellular amount of CELF2,

and found that LPS + IL10 treatment significantly increases CELF2 binding to pre-miR155 (Fig 3).

To assess the contribution of CELF2 to IL10 function we generated CELF2 knockdown cells using CRISPR-Cas9 and guide RNA (gRNA) mediated silencing. CELF2 and non-CELF2 knockdown cells were stimulated with LPS +/- the indicated concentrations of IL10. As Fig 4 shows, CELF2 knockdown impaired IL10's ability to inhibit miR155 expression, but this inhibition was partial suggesting that CELF2's role is partly dispensable or a compensatory mechanism exists for the loss of CELF2. We then examined the effect of CELF2 knockdown on IL10's ability to inhibit LPS-induced TNFα production since miR155 is involved in LPS stimulated TNFα expression [16]. IL10 at low concentrations (0.5 ng/ml) could inhibit TNFα expression in non-target knockdown cells, but not in CELF2 knockdown cells. However, at high IL10 concentrations (10 ng/ml) IL10 could partly inhibit TNFα expression, again suggesting a compensatory mechanism coming into play. One potential compensating mechanism may be the contribution of the highly related homologue, CELF1 [48].

To identify RNA binding proteins that contribute to miRNA processing, Treiber *et al*. performed a screen for binders of immobilized pre-miRNA hairpins [49]. CELF1 and CELF2 were isolated as proteins which bind to the basal stem of pre-miR140, and they showed that expression of either CELF1 or CELF2 in cells inhibits miR140 expression. Deletion of either allows some miR140 expression, and deletion of both CELF1 and CELF2 is required for full miR140 expression [49]. Treiber *et al*. suggest that CELF1/2 binding to the pre-miR140 basal stem might interfere with DICER binding to pre-miR140 and thus inhibit its processing to miR140. Treiber *et al*. did not find miR155 binding to either CELF1 or CELF2, but they also did not observe KSRP binding to miRNAs including miR155, and they write that their *in vitro* miRNA-protein binding screen may give different results than cell-based binding assays [49]. They also note the possibility of cell specific type differences. In their studies theyused human cancer lines, none of which were macrophage or monocyte in origin.

The binding of CELF2 to pre-miR155 might also inhibit the ability of DICER to bind to and cleave pre-miR155. Alternatively, CELF2 binding to pre-miR155 could prevent the action of other proteins needed for pre-miR155 processing such as KSRP [16]. Studies are ongoing to determine whether both CELF1 and CELF2 bind to pre-miR155 in IL10 treated cells, and whether this is the only mechanism by which IL10 inhibits miR155 expression.

## Supporting information

**S1 Fig.**
(TIF)

**S1 Raw images.**
(PDF)

## Acknowledgments

We thank Jenny Moon and Dr. Leonard Foster of UBC Proteomics Core Facility for their guidance and help with mass spectrometry.

## Author Contributions

**Conceptualization:** Sylvia T. Cheung, Alice L-F. Mui.

**Data curation:** Jeff S. J. Yoon, Mike K. Wu, Tian Hao Zhu, Helen Zhao, Sylvia T. Cheung, Thomas C. Chamberlain, Alice L-F. Mui.

**Formal analysis:** Jeff S. J. Yoon, Mike K. Wu, Alice L-F. Mui.

**Funding acquisition:** Alice L-F. Mui.

**Investigation:** Jeff S. J. Yoon, Mike K. Wu, Tian Hao Zhu, Helen Zhao, Thomas C. Chamberlain, Alice L-F. Mui.

**Methodology:** Jeff S. J. Yoon, Mike K. Wu, Sylvia T. Cheung, Alice L-F. Mui.

**Project administration:** Alice L-F. Mui.

**Resources:** Alice L-F. Mui.

**Supervision:** Thomas C. Chamberlain, Alice L-F. Mui.

**Validation:** Jeff S. J. Yoon, Mike K. Wu, Sylvia T. Cheung, Thomas C. Chamberlain, Alice L-F. Mui.

**Visualization:** Jeff S. J. Yoon, Mike K. Wu, Thomas C. Chamberlain, Alice L-F. Mui.

**Writing – original draft:** Jeff S. J. Yoon, Mike K. Wu, Thomas C. Chamberlain, Alice L-F. Mui.

**Writing – review & editing:** Jeff S. J. Yoon, Mike K. Wu, Thomas C. Chamberlain, Alice L-F. Mui.

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
