## [Decision Letter · Decision Letter 0]

18 Mar 2020

PONE-D-19-31111

Interleukin-10 control of pre-miR155 maturation involves CELF2

PLOS ONE

Dear Dr Mui,

Thank you for submitting your manuscript to PLOS ONE. After careful consideration, we feel that it has merit but does not fully meet PLOS ONE’s publication criteria as it currently stands. Therefore, we invite you to submit a revised version of the manuscript that addresses the points raised during the review process. Specifically, Authors should provide all of the mass spectrometry data as there are many additional proteins that are also binding to pre-miR155. Moreover, Authors also should quantitate the western blots and provide high resolution versions of alll the images depicted in the mansucript.

We would appreciate receiving your revised manuscript by May 02 2020 11:59PM. To enhance the reproducibility of your results, we recommend that if applicable you deposit your laboratory protocols in protocols.io, where a protocol can be assigned its own identifier (DOI) such that it can be cited independently in the future. For instructions see: http://journals.plos.org/plosone/s/submission-guidelines#loc-laboratory-protocols

We look forward to receiving your revised manuscript.

Kind regards,

Fabrizio Mattei, Ph.D.

Academic Editor

PLOS ONE

Journal Requirements:

Reviewers' comments:

Reviewer's Responses to Questions

**Comments to the Author**

1. Is the manuscript technically sound, and do the data support the conclusions?

Reviewer #1: Yes

Reviewer #2: Yes

2. Has the statistical analysis been performed appropriately and rigorously? 

Reviewer #1: Yes

Reviewer #2: Yes

3. Have the authors made all data underlying the findings in their manuscript fully available?

Reviewer #1: No

Reviewer #2: Yes

4. Is the manuscript presented in an intelligible fashion and written in standard English?

Reviewer #1: Yes

Reviewer #2: Yes

5. Review Comments to the Author

Reviewer #1: This concise study demonstrates that CELF2 binds to pre-miR155 and is important to the proper processing of pre-miR155 and regulation of TNFalpha. The data adds to the growing list of proteins that have been implicated in miR155 regulation and supports the authors conclusions. However, the authors should provide all of the mass spec data as there are many additional proteins that are also binding to pre-miR155 and this is important information for the community as well as being appropriate practice when a proteomics study is performed.

Reviewer #2: The authors Yoon et al. have very elegantly described the regulation of miR-155 biogenesis by LPS and IL-10 and have identified a novel interaction with CELF2 and pre-miR-155.

Minor comments,

Please provide high resolution images for all figures.

The authors should quantitate the western blots.

Line 408: TPP should be TTP.

6. PLOS authors have the option to publish the peer review history of their article (what does this mean?). If published, this will include your full peer review and any attached files.

Reviewer #1: No

Reviewer #2: No

---

## [Author Response · Author response to Decision Letter 0]

24 Mar 2020

As requested by the Editor, we have provided original uncropped and unadjusted images underlying all blot results reported in our figures. These images can be found within the file S1_raw_images.pdf within Supporting Information.

Reviewer #1: This concise study demonstrates that CELF2 binds to pre-miR155 and is important to the proper processing of pre-miR155 and regulation of TNFalpha. The data adds to the growing list of proteins that have been implicated in miR155 regulation and supports the authors conclusions. However, the authors should provide all of the mass spec data as there are many additional proteins that are also binding to pre-miR155 and this is important information for the community as well as being appropriate practice when a proteomics study is performed.

Answer: 

The mass spectrometry proteomics data has been deposited to ProteomeXchange via PRIDE database with the title “Identification of pre-miR155 binding proteins from macrophages treated with LPS or LPS + IL10”. The dataset is currently private, but can be accessed with the following reviewer account: 

Username: reviewer59427@ebu.ac.uk

Password: IrTzQ6rb

Identifier: PXD015750

The dataset will be publicly released later when the manuscript is published.

The information about the dataset and the database has been added to the “Acknowledgement” and “Method” section of the manuscript.

Reviewer #2: The authors Yoon et al. have very elegantly described the regulation of miR-155 biogenesis by LPS and IL-10 and have identified a novel interaction with CELF2 and pre-miR-155.

Minor comments,

Please provide high resolution images for all figures.

The authors should quantitate the western blots.

Line 408: TPP should be TTP.

Answers: 

Thank you for pointing out the misspelling of TTP protein. The misspelling has been corrected in the revised manuscript.

For the immunoblot images presented in the manuscript, all proteins were quantified and presented as graphs with the exception of the KSRP protein in pulldown (Fig. 1C). 

We now provide the quantification in graphs for KSRP in pulldown as Supplementary Figure 1 since KSRP is a control not central to the paper. If requested by the reviewer, we can include the KSRP quantification graph in main Figure 1. A summary of protein quantification graph locations is summarized in Table 1 below.

Table 1 – Description of immunoblot images and corresponding graphs 

Immunoblot images Protein Location of quantification graphs

Figure 1A TTP Figure 1A

 KSRP Figure 1A

Figure 1B TTP Figure 1C, 1D

 KSR Supplementary Figure 1

Figure 3A CELF2 Figure 3A

Figure 3B CELF2 Figure 3C, 3D

Figure 4A CELF2 Figure 4A

High-resolution images of the figure in TIF file format are available via links in the upper right corner of the figure images in manuscript PDF file. 

Or use the following links below:

Figure 1: 

https://www.editorialmanager.com/pone/download.aspx?id=25477811&guid=349534b7-0af4-4470-96c6-aaaa94da1fa3&scheme=1

Figure 2: 

https://www.editorialmanager.com/pone/download.aspx?id=25477812&guid=ac1b0815-6d04-4a52-81b1-e83ba7a8f311&scheme=1

Figure 3: 

https://www.editorialmanager.com/pone/download.aspx?id=25477813&guid=2cf84fbd-6aa1-4a05-a565-e618f2d5336d&scheme=1

Figure 4: 

https://www.editorialmanager.com/pone/download.aspx?id=25477814&guid=ed28a848-5847-408e-92b3-626d57b74ac2&scheme=1

Figure 5:

https://www.editorialmanager.com/pone/download.aspx?id=25477815&guid=fb43e883-52a5-4e83-955c-63174c11296c&scheme=1

---

## [Decision Letter · Decision Letter 1]

30 Mar 2020

Interleukin-10 control of pre-miR155 maturation involves CELF2

PONE-D-19-31111R1

Dear Dr. Mui,

We are pleased to inform you that your manuscript has been judged scientifically suitable for publication and will be formally accepted for publication once it complies with all outstanding technical requirements.

With kind regards,

Fabrizio Mattei, Ph.D.

Academic Editor

PLOS ONE

Additional Editor Comments (optional):

Reviewers' comments:

Reviewer's Responses to Questions

**Comments to the Author**

1. If the authors have adequately addressed your comments raised in a previous round of review and you feel that this manuscript is now acceptable for publication, you may indicate that here to bypass the “Comments to the Author” section, enter your conflict of interest statement in the “Confidential to Editor” section, and submit your "Accept" recommendation.

Reviewer #1: All comments have been addressed

Reviewer #2: All comments have been addressed

2. Is the manuscript technically sound, and do the data support the conclusions?

Reviewer #1: Yes

Reviewer #2: Yes

3. Has the statistical analysis been performed appropriately and rigorously? 

Reviewer #1: Yes

Reviewer #2: Yes

4. Have the authors made all data underlying the findings in their manuscript fully available?

Reviewer #1: Yes

Reviewer #2: Yes

5. Is the manuscript presented in an intelligible fashion and written in standard English?

Reviewer #1: Yes

Reviewer #2: Yes

6. Review Comments to the Author

Reviewer #1: (No Response)

Reviewer #2: The authors have appropriately addressed all the concerns raised by the reviewers and have made the revisions/corrections requested.

7. PLOS authors have the option to publish the peer review history of their article (what does this mean?). If published, this will include your full peer review and any attached files.

Reviewer #1: No

Reviewer #2: No

---

## [Editor Report · Acceptance letter]

10 Apr 2020

PONE-D-19-31111R1 

Interleukin-10 control of pre-miR155 maturation involves CELF2 

Dear Dr. Mui:

I am pleased to inform you that your manuscript has been deemed suitable for publication in PLOS ONE. Congratulations! Your manuscript is now with our production department. 

With kind regards,

on behalf of

Dr. Fabrizio Mattei 

Academic Editor

PLOS ONE